# New Safety Aspects in Corneal Donation—Studies on SARS-CoV-2-Positive Corneal Donors

**DOI:** 10.3390/jcm11123312

**Published:** 2022-06-09

**Authors:** Diana Wille, Joana Heinzelmann, Astrid Kehlen, Marc Lütgehetmann, Dominik S. Nörz, Udo Siebolts, Anke Mueller, Matthias Karrasch, Nicola Hofmann, Anja Viestenz, Martin Börgel, Ferenc Kuhn, Arne Viestenz

**Affiliations:** 1Department of Ophthalmology, Halle, University Hospital Halle-Wittenberg, 06120 Halle, Germany; anja.viestenz@uk-halle.de (A.V.); fkuhn@mindspring.com (F.K.); arne.viestenz@uk-halle.de (A.V.); 2Mitteldeutsche Cornebank Halle, University Hospital Halle-Wittenberg, 06120 Halle, Germany; 3German Society for Tissue Transplantation (DGFG) gGmbH, 30625 Hannover, Germany; nicola.hofmann@gewebenetzwerk.de (N.H.); martin.boergel@gewebenetzwerk.de (M.B.); 4Department for Laboratory Medicine, Unit III, Molecular Diagnostic Section, University Hospital Halle-Wittenberg, 06120 Halle, Germany; astrid.kehlen@uk-halle.de (A.K.); matthias.karrasch@uk-halle.de (M.K.); 5Institute for Medical Microbiology, Virology and Hygiene, University Hospital Hamburg-Eppendorf, 20246 Hamburg, Germany; mluetgeh@uke.de (M.L.); d.noerz@uke.de (D.S.N.); 6Institute of Pathology, University Hospital Halle-Wittenberg, 06112 Halle, Germany; udo.siebolts@uk-koeln.de; 7Institute for Medical Microbiology, University Hospital Halle-Wittenberg, 06112 Halle, Germany; anke.mueller3@uk-halle.de; 8Helen Keller Foundation for Research and Education, Birmingham, AL 35233, USA; 9Department of Ophthalmology, University of Pécs Medical School, 7632 Pécs, Hungary

**Keywords:** cornea, donation, cornea transplant, corneal tissue, qRT–PCR, subgenomic RNA, immunohistochemistry, SARS-CoV-2

## Abstract

In the tissue donation field, to prevent pathogen transmission, all donors are screened by postmortem swabs for SARS-CoV-2 using qRT–PCR. Corneas from donors who tested positive for SARS-CoV-2 were subjected to further investigations. Corneal transplants and culture medium from positive donors were cultured under appropriate safety conditions for further analyses. Cornea tissue samples, including sclera/limbus/cornea, and culture media were taken at different time points for testing for SARS-CoV-2 using qRT–PCR, immunohistochemistry (IHC) and subgenomic RNA (sgRNA) analysis. Between January and May 2021, in four donors with initial negative premortem rapid tests, SARS-CoV-2 was detected post-mortem using qRT–PCR. In these cases, SARS-CoV-2 was observed at the beginning of cultivation in both tissue and culture medium using qRT–PCR and IHC. The virus was mainly localized in the limbus epithelial cells, with a stable detection level. Premortem rapid tests are potentially insufficient to exclude SARS-CoV-2 infection in corneal donors. While, for SARS-CoV-2, the risk of infection via transplants is considered low, a residual risk remains for presymptomatic new infections. However, our investigations provide the first indications that, with organ cultures, the risk of virus transmission is minimized due to the longer minimum culture period.

## 1. Introduction

Coronavirus disease 19 (COVID-19) is caused by the novel betacoronavirus identified as severe acute respiratory syndrome coronavirus 2 (SARS-CoV-2). The virus spread from Wuhan, China, in December 2019 and rapidly caused a worldwide pandemic, resulting in major changes in virtually all aspects of life.

In the field of tissue donation, it is of particular importance to prevent the transmission of pathogens for the safety of recipients. In addition to testing donors for possible viral diseases, the collection of an extended medical history is an important part of the donor-screening process and is required by the German Federal Institute for Vaccines and Biomedicines (Paul-Ehrlich-Institute). All donors are strictly assessed regarding the risk they may pose; additionally, they are tested at a donation headquarters for SARS-CoV-2 using pooled nasopharyngeal-conjunctival swabs by real-time quantitative reverse transcription polymerase chain reaction (qRT–PCR).

The tests to be applied on living patients are based on qRT–PCR, enzyme-linked immunosorbent assay (ELISA) or lateral-flow assays to detect the virus in respiratory material. Seroconversion occurs 6–12 days after the beginning of symptoms and can be detected in the serum or plasma of patients [1].

Currently, the risk of virus transfer by a transplanted cornea or other ophthalmologic tissue is unclear. Although no case of transfer of the virus during tissue transplantation has been reported, the data regarding risk are not without controversy.

While various studies have shown the presence of SARS-CoV-2 in conjunctival swab samples, aqueous humor and vitreous fluid [2,3], the results from other studies are conflicting [4]. Subgenomic RNA (sgRNA), which can provide an indication of the replicative capacity of the virus, has been detected in cornea samples, but virus isolation from Vero cells infected with tissue samples failed [2,5], so no conclusion can be drawn regarding whether the detection of sgRNA indicates replication-capable viruses. Therefore, for the safety of tissue recipients, the infection status of donors is of particular importance.

Previous studies about SARS-CoV-2 infections in ocular tissues were mostly performed using ocular tissues from patients who died on SARS-CoV-2. In contrast, in this study, we investigated the presence of the SARS-CoV-2 virus in ocular tissue from presumed negative-tested cornea donors with post-mortem verified COVID-19 infection. Ocular tissue was analyzed using qRT–PCR, subgenomic RNA analyses and immunohistochemical stainings. Thereby, we monitored the local and time-related distribution of the SARS-CoV-2 virus during regular organ culture of corneas. Thus, the study presented here aims to determine whether there is an increased risk of transmission of SARS-CoV-2 from cornea donors.

## 2. Materials and Methods

### 2.1. Human Sample Preparation and Cultivation

In 2020, 165 donors were tested for SARS-CoV-2, without any positive results. In 2021, 106 donors were tested for SARS-CoV-2 in the period described, and 4 were positive.

Eight corneas of four patients were studied in this report. All patients were treated at a collaborating hospital, and they all initially tested negative for SARS-CoV-2 using a lateral flow test. After death, the patients’ relatives consented to donating the corneas. During donation, the periocular area was disinfected with 1.5% PVP-iodine solution for 3 min.

After the eye was enucleated, the preparation of the corneoscleral disk was accomplished following a repeat disinfection of the globe with a 2% PVP-iodine solution in the cornea bank. Corneoscleral disk cultivation at 37 °C and in 5% (*v*/*v*) CO_2_ was performed in “Culture Medium 1” (Biochrom Berlin); a regular medium change took place between the 3rd and 7th day of culturing. As part of the donation process, nasopharyngeal and conjunctival swabs were taken before disinfection and tested by pooling for SARS-CoV-2 by qRT–PCR.

The available donor characteristics are listed in Table 1.

With informed consent of the relatives, for the positive 4 cases, both donor corneas with SARS-CoV-2-positive results were further cultured in organ culture and treated under quarantine conditions for the analysis.

Corneas were divided into circle segments using a scalpel to allow time-dependent detection of the virus in the cornea during cultivation time using qRT–PCR, sgRNA and IHC analyses (Figure 1).

### 2.2. Test Kits Used for SARS-CoV-2 Detection

For qRT–PCR, an “Anchor SARS-CoV-2 PCR Kit” (Anchor Diagnostics GmbH, Hamburg, Germany), “Xpert ^®^ X-Press SARS-CoV-2” (Cepehid Inc., Sunnyvale, CA, USA) and “Alinity m SARS-CoV-2 Kit” (Abbott Molecular Inc., Des Plaines, IL, USA) were used. For the preparation of tissue samples, an “RTP DNA/RNA Virus Mini Kit” (Stratec Molecular GmbH, Berlin, Germany) and “RTP Pathogen Kit” (Invitek Molecular GmbH, Berlin, Germany (case 4)) were used. For antibody tests, an “Anti-SARS-CoV-2-Elisa” (Euroimmun AG, Lübeck, Germany) and “SARS-CoV-2 IgG-Assay” (Abbott Molecular Inc., Des Plaines, IL, USA) were used. All tests were performed according to the manufacturers’ instructions.

### 2.3. Subgenomic RNA Analysis

Quantitative detection of the SARS-CoV-2 N-gene compared to the sgRNA-N-gene was performed as previously described [2,6]. Briefly, CDC-N1 assay primers and probes [7] were used in conjunction with a 5′-UTR primer (leader sequence) to exclusively detect and compare levels of subgenomic-N. SARS-CoV-2 (HH-1)-infected Vero cell culture supernatant served as a control [8]. Amplification and detection were carried out on a LightCycler 480 instrument (Roche Diagnostics, Basel, Switzerland) using a Roche RNA process control kit mastermix (Roche Diagnostics, Basel, Switzerland).

### 2.4. Immunohistochemistry

At days 2, 7 and 28, a circle segment of the corneal grafts was used for IHC staining. As control tissue, an enucleated eye from COVID-19-negative patient was added to the analyses. The tissues were carefully washed using phosphate-buffered saline, fixed by incubating in 4% paraformaldehyde and embedded in paraffin. Serial sections (4 µm) were prepared for IHC staining. Paraffin sections were incubated at 60 °C for 2 h and then dehydrated with xylene, followed by decreasing ethanol concentrations and finally distilled water. Antigen retrieval was performed using an EDTA buffer (pH 9) for 45 min in a water bath at 95 °C. The endogenous peroxidase activity of the sections was blocked by using 3% H_2_O_2_ for 10 min. After blocking, background tissue sections were incubated with a primary antibody (anti-SARS-CoV-2 monoclonal mouse antibody, 1:50, BSB-3701-01, Medac, Wedel, Germany) for 24 h at 4 °C, followed by washing steps and incubation with an Alexa Fluor 488-conjugated secondary antibody (anti-mouse IgG, 4409S, Cell Signalling, Danvers, MA, USA). Sections were counterstained and mounted with Prolong Gold Antifade Mountant (ThermoFisher Scientific, Waltham, MA, USA). SARS-CoV-2 expression was evaluated by estimating a semiquantitative scoring system (light staining (+), moderate staining (++) and strong staining (***)). Evaluation was done by two experienced biologists.

## 3. Results

Of the 165 donations tested in 2020, none had a positive signal in the tested postmortem swab. Of the 106 donations tested in 2021, four donors had a positive postmortem qRT–PCR test for SARS-CoV-2 virus RNA, despite extensive donor history and negative screening tests at the treating hospitals, using the procedure described.

### 3.1. Case History

In these four cases (Table 1), an attempt was made to determine the route of infection and, if possible, the infection status based on available data and research.

In case 1, due to previous recurrent bacterial infections and SARS-CoV-2 evidence with a high CT value, there was probably a persistent infection with SARS-CoV-2 in an immunocompromised patient. In a previous hospitalization, four to six weeks before death, the patient tested several times positive (CT values 30–33) and negative for SARS-CoV-2 by qRT–PCR. At the hospital, where the patient died on the same day, he tested negative on admission using a lateral flow assay.

In cases 2 and 3, the patients tested negative on admission to the hospital. Based on the length of hospital stay and the CT values of the swabs, we assumed a new infection, and the patients were admitted prior to the beginning of symptoms. Case 2 tested negative several times by qRT–PCR before transfer to the hospital. During the three-weeks hospital stay, he tested negative for SARS-CoV-2 several times by lateral flow assay. Case 3 tested negative by qRT-PCR on admission to hospital and once during the eight-day length of stay.

In case 4, the high CT value (at the detection limit) in combination with the missing antibodies, could indicate a very low viral load or a longer past infection. A possible route of infection could not be determined.

Case 1: persistent infection, immunocompromised patient

Case 2/Case 3: pre-symptomatic new infections

Case 4: very low viral load/longer past infection

### 3.2. SARS-CoV-2 Detection in Corneal Tissue and Related Culture Medium Using qRT–PCR

In cases 1–3 (CT values of donor swabs <30), SARS-CoV-2 RNA was detected in the culture medium before medium change. In case 4 (CT value of the donor swab >35), no SARS-CoV-2 RNA was detected. After regular medium change, SARS-CoV-2 RNA was detected once (case 3/14 d) in the new culture medium. In the corneal grafts, SARS-CoV-2 RNA could be detected for case 2 and case 3 over the complete culture period, with stable CT values. The division of the corneal circle segments showed that a higher proportion of SARS-CoV-2 RNA was detectable in the limbal–scleral area than in the central corneal area. For case 4, no SARS-CoV-2 RNA could be detected in the corneal tissue, so cultivation was not continued after day 7 (see Table 2).

### 3.3. Subgenomic RNA Analysis

Due to SARS-CoV-2 positivity in cases 2 and 3, the samples were additionally analyzed for the presence of sgRNA.

For case 2, sgRNA (sg N1) was detected in both corneal graft segments (right and left) at day 7. The CT value was higher than that of genomic RNA (N1), and the delta Ct was between 3.4 and 4.1. On day 21, the CT values for samples of both corneal segments, for both genomic and sgRNA, were below the detection limit. The segments were divided into limbal–scleral and central parts. In the limbal–scleral area, in contrast to the central area, genomic and sgRNA could be detected in the distribution on day 7 (Table 3).

In case 3, genomic RNA (N1) and sgRNA (sg N1) were also detected in the corneal segments of both corneas in the day 7 samples. After the division of both corneas into limbal–scleral and central areas, only the limbal–scleral area, but not the central area, was positive for SARS-CoV-2. As before, the CT value of the sgRNA was higher than that of the genomic RNA. At day 14, the detection of sgRNA was performed in the whole segment samples of both corneas; at day 21, it was performed only in the segment sample of the right cornea (Table 4).

### 3.4. Immunohistochemical Staining for SARS-CoV-2

For case 2, samples of both the left and right corneal circle segments were immunohistochemically stained for SARS-CoV-2 nucleocapsid after 7 days and 32 days of cultivation. As a negative control, corneal tissue from a SARS-CoV-2-negative enucleated eye was used. IHC analyses showed no SARS-CoV-2-positive cells in the control (data not shown). In contrast, cytoplasmic SARS-CoV-2 staining was revealed in the epithelial cells of the right and left eye corneal circle segments after 7 days of cell culture (Figure 2).

In the right cornea, the basal epithelial cells of the limbus (limbus epithelial stem cells) and the corneal epithelial cells showed stronger cytoplasmic SARS-CoV-2-staining than conjunctival epithelial cells. In the left cornea, the circular segment conjunctival epithelial cells and basal limbus epithelial cells showed stronger cytoplasmic staining than corneal epithelial cells. At day 32, in the circular segments of both the right and left corneas, the SARS-CoV-2 antigen was detected in epithelial cells (Figure 3).

In the right eye, mainly the epithelial cells of the cornea and limbus were positive for SARS-CoV-2, whereas in the left eye, this effect was seen predominantly in the epithelial cells of the conjunctiva. On day 32, the strongest intracellular signal was identified in the nucleus of the epithelial cells.

The overview of the immunohistochemical results can be found in Table 5.

## 4. Discussion

Typically, donors of corneal tissue die in a hospital. Upon admission and stay, they undergo strict testing and are observed for various symptoms. Despite all this screening, presymptomatic new infections remain a potential risk, which can never be reduced to zero. To date, no case of SARS-CoV-2 transmission via transplantation of infected corneal tissue has been described, and the general risk is considered low [9,10]. In the coronavirus pandemic era, the question arises whether existing requirements for transplant tissue safety are sufficient and whether further testing would increase the efficiency of the current safety measures.

As of March 2020, all donors for the Mitteldeutsche Corneabank Halle (MCH) were tested for SARS-CoV-2 by a pooled nasopharyngeal-conjunctival swab. Therefore, one positive donor was identified by a cooperating hospital, without detection in the postmortem swab, culture medium or corneal tissue [11]. A survey of the German cornea banks by the German Ophthalmological Society (DOG) showed that out of 26 cornea banks surveyed, eight banks test swabs as additional samples from the donors. In 2020, no swab was found to be positive among the 200 tested donors [12]. In 2020, the patients were tested by qRT–PCR upon admission to the hospital. Starting in 2021, the test methods for hospital admission were gradually changed to the less complex lateral flow assays. From January to May 2021, we, at the MCH, detected SARS-CoV-2 in the pooled swab of four donors, although the lateral flow assay results were negative for all four patients. Due to the limited data available regarding the transmissibility of the virus, we decided to further investigate the tissues after obtaining the consent of the relatives. For this reason, the corneas were also cultured as regular cornea grafts in dedicated laboratories.

Regarding the four described donors, in case 1, the patient had a known past SARS-CoV-2 infection; in cases 2 and 3, the patients had presymptomatic new infections; in case 4, the time of infection was unknown. In case 4, the positive swab test could not be confirmed with tissue or culture medium of corneal grafts via qRT–PCR. qRT–PCR analyses of cultivated corneal graft tissue identified SARS-CoV-2 RNA for 2 of the 4 described cases. Both SARS-CoV-2-positive corneal grafts were from non- or presymptomatic donors. The CT values of the tissues for cases 2 and 3 remained stable over the complete culture period, 32.4–36.3 and 30.6–36.5, respectively.

Analyzing culture media using qRT–PCR at different time points, we were able to identify viral RNA from presymptomatic donors and from post-infection donors, even when the viral RNA was still at a low concentration. In the culture medium of case 2, SARS-CoV-2 RNA could not be detected, as the medium change had already been carried out; for case 3, SARS-CoV-2 detection was positive in the right cornea 7 days after the medium change (day 14). The fact that SARS-CoV-2 RNA was also detectable 14 days after the medium change in case 3 may indicate that the virus in corneal grafts can be active for an extended time. In donors with a CT value below 30, the detection of SARS-CoV-2 in the culture medium of cornea graft cultures before a medium change has been recently confirmed by Thaler et al. [13].

SARS-CoV-2 utilizes angiotensin-converting enzyme 2 (ACE2) and the membrane-bound serine protease TMPRSS2 for host cell entry [14] Therefore, potential infection of tissues is linked to the expression of these enzymes. While some publications describe negligible ACE2 expression in the conjunctiva, concluding that infection via this pathway is unlikely [15], other studies have reported detection of these enzymes in the cornea and conjunctiva [16]. Studies have shown that the highest expression of ACE with TMPRSS2 is detected in nasal secretory cells, which correlates with COVID-19 pathology. In the cornea, ACE2 and TMPRSS2 expression was detected in the limbal and conjunctival epithelium [17,18]. Since the literature available thus far contains contrasting statements on the presence of the receptors required for binding the virus in the different tissues [16,19], the limbal–scleral area and the corneal area were separately investigated for virus material. Therefore, circular segments of the cornea grafts from the presymptomatic cases were divided again and analyzed at different time points. In the limbus/sclera tissue of both cornea grafts for cases 2 and 3, the genomic RNA from the virus was verified at all investigation time points. Crucially, in the corneal tissue, SARS-CoV-2 RNA was either not detectable or was present at a level close to the detection threshold. These results provide evidence that cells of the limbus and sclera are more readily infected by SARS-CoV-2 than corneal cells.

Since the detection of SARS-CoV-2 RNA by qRT–PCR does not necessarily mean that the virus is active and, thus, that the transplanted tissue is potentially infectious, samples from cases 2 and 3 were tested for sgRNA during the cultivation period. SgRNA is produced only in the replication cycle of the virus. However, sgRNA is detectable even several time after replication, because its location in double membrane vesicles provides some protection against the degradation processes. Based on this, it can be concluded that replication occurred, which was accompanied by the formation of the sgRNA, but that does not necessarily mean that replication is currently occurring [20]. In both cases, sgRNA was detectable in the cornea graft of SARS-CoV-2-positive donors during the cultivation time, but only in the limbal–scleral tissues. These results suggest that SARS-CoV-2 replication has occurred in the limbus–scleral region but not that it is still occurring. Studies have demonstrated that the expression of sgRNA is significantly suppressed in asymptomatic patients and that the ratio of sgRNA to genomic RNA correlates with disease severity [21], but other studies have refuted this [22]. Therefore, the ratio of sgRNA to genomic RNA of our tested samples cannot be used to infer the disease severity or potential current infection status. Our samples were not suitable for infecting Vero cells, and evidence of SARS-CoV-2 replication from corneal tissue has also not been described to date [2,5].

Using IHC of SARS-CoV-2 for case 2 on day 7, we showed that in the cornea, mainly epithelial cells were infected. In the limbus, strong cytoplasmic SARS-CoV-2 antigen staining was found in the basal epithelial cells (probably limbal epithelial stem cells) of both grafts, changing to strong staining of the nucleus towards the end of cultivation. These results indicate that for SARS-CoV-2, the primary targets are limbal epithelial cells. Limbal stem cells are the source of cell supply for the centripetally renewing corneal epithelium. Interestingly, the detection of SARS-CoV-2 based on the nucleocapsid protein using immunohistochemical staining strongly supports this hypothesis of decreased viral activity during the period of cultivation, in that the intracellular location changed during the cultivation time. On day 7, the nucleocapsid protein was detected in the cytosol of epithelial cells of corneal grafts. On day 32, immunohistochemical staining revealed strong nuclear staining of the nucleocapsid protein in the epithelial cells. The nucleocapsid protein is the main component of viral structure proteins and the most abundant protein in virus-infected cells. The nucleocapsid protein plays an important role in the process of virus infection, replication and packaging. Recent studies have confirmed that the nucleocapsid protein of coronavirus not only is localized in the cytosol of host cells, but that it also translocates into the nucleus [23]. By entering the nucleus, the nucleocapsid protein binds to the proteasome activator PA28γ as part of the 20S proteasome and is then degraded internally under the action of the 20S proteasome [24]. It has been described that by degrading the coronavirus nucleocapsid protein in the nucleus of host cells, PA28γ plays a major role in viral infection and pathogenesis [23,25,26]. The nuclear detection of SARS-CoV-2 nucleocapsid proteins in corneal grafts after 32 days could indicate late phases of infections with high concentrations of degraded SARS-CoV-2 nucleocapsid protein and thereby decreased viral replication and latency.

For organ cultures, most corneal grafts are transplanted between 14 and 21 days after removal, so this time frame must be considered critical. To the best of our knowledge, no information is available regarding whether corneal grafts are potentially infectious when SARS-CoV-2 infection may be evidenced in the tissue and culture medium, even with high CT values. No statement can be made based on CT values as to whether this is potentially infectious material, as there are no comparative values, similar to respiratory material [1]. Although many studies have examined tissues from deceased patients with known SARS-CoV-2 infection, these data cannot be applied directly to corneal grafting [2]. Studies of living patients [27] show detection of the N-Protein in conjunctiva, trabecula and iris cells. The nasolacrimal system, as an anatomical bridge between the ocular surface and the nasolacrimal duct, has been discussed as a possible entry route [28,29,30]. In the present study, SARS-CoV-2 infections were incidental findings, whose data could not otherwise have been examined, as previous studies have examined patients with longer-known SARS-CoV-2 infection, and we included two cases of probably presymptomatic new infections.

The results of detection of SARS-CoV-2 by PCR and the sgRNA analysis and the immunohistochemical evidence regarding the fixed corneas indicate that infection of the corneas with SARS-CoV-2 may occur via the limbal–scleral region. Limbal stem cells produce the superficial corneal epithelium and provide a barrier to the conjunctiva. SARS-CoV-2 utilizes ACE2 and the membrane-bound serine protease TMPRSS2 for host cell entry [14]. Therefore, the potential infection of tissues is linked to the expression of these enzymes. Whether post-transplant infection is possible via the naso-lacrimal system cannot be ascertained on the basis of the results, but the risk of infection also cannot be completely excluded. These findings are not only relevant in terms of the suitability of COVID-positive donor corneas for keratoplastics, but also important for limbocorneal grafts and sclerocorneal grafts, because the cultured limbal region may contain active SARS-CoV2-virus. Long-term cultivation of corneas in organ culture may have an effect on virus replication. In this context, organ culture and other methods with a shorter cultivation period should be differentiated in the risk assessment with regard to virus transmission.

## 5. Conclusions

First, because the change in hospital testing strategy directly coincides with previously undetected SARS-CoV-2 infections, this issue should be critically considered as a safety factor. Previously published studies have indicated that infections that have already occurred carry a low risk of transmission [2]. In the case of new infections, infectivity cannot be excluded, and an increased risk remains with these donors.

Second, we were able to confirm the presence of SARS-CoV-2 in culture medium before changing the medium, even after a previous infection, if the CT value of the donor swab was in a critical range (CT value < 30). Should this finding be confirmed in further investigations, an additional test method would be available for testing for SARS-CoV-2 in the tissue processing area.

Third, both the qRT–PCR studies and the data from the sgRNA and IHC studies pointed to the limbus–scleral region as a possible site of virus entry and replication. Although the formation of the corneal epithelium by the limbus cannot exclude transmission of SARS-CoV-2, prolonging the duration of the organ culture can affect virus replication.

Fourth, changes in hospital testing strategies after 2020 have increased the potential risk of undetected reinfection of potential donors with SARS-CoV-2. Donors with a presymptomatic new infection represent a risk for transmissibility. To estimate the rate of new infections among donors, the data of the testing banks should be evaluated closely, and if necessary, an expansion of the donor screening activities should be considered.

Our study’s obvious limitations include its small size and the different sampling points, which allow only limited comparability. Nevertheless, we think that we identified a possible route of virus transmission and recommend further investigations to verify this finding so that the risk of transmission via corneal grafting be eliminated.

## Figures and Tables

**Figure 1 jcm-11-03312-f001:**
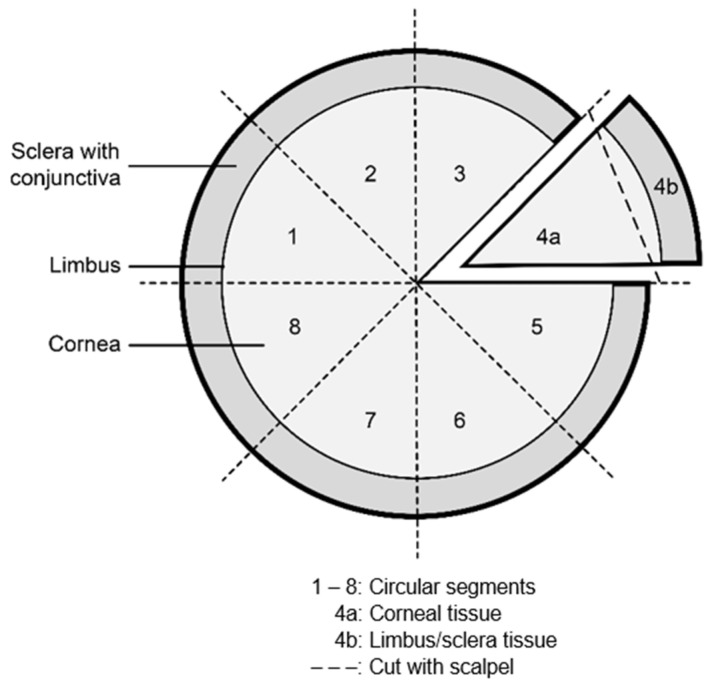
Overview of cornea graft preparation in circle segments.

**Figure 2 jcm-11-03312-f002:**
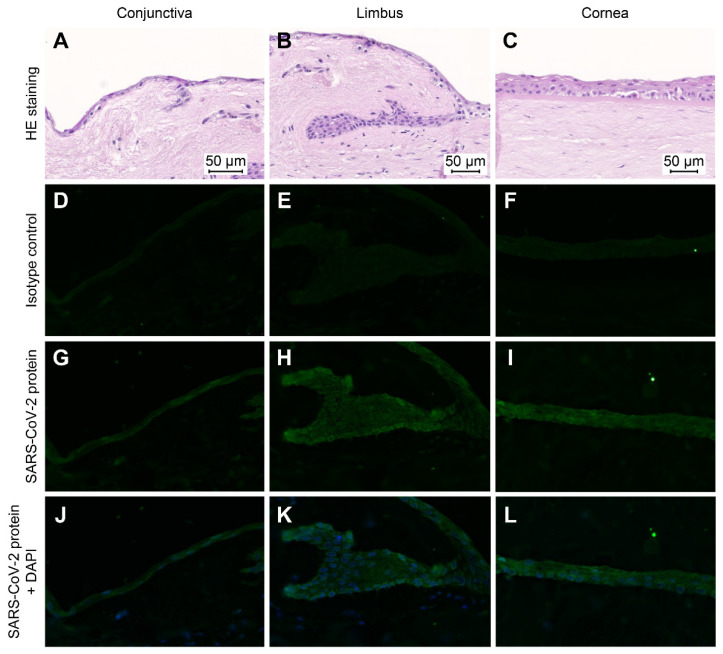
Immunohistochemical staining (Case 2, day 7). The SARS-CoV-2 Nucleocapsid protein was detected in the right cornea tissue of the COVID-19 donor (case 2) after 7 days of in vitro cultivation. HE-staining of ocular tissues from the conjunctiva, limbus and cornea (**A**–**C**). Isotype control of immunofluorescence analyses of ocular tissues (**D**–**F**). Immunofluorescence imaging of ocular tissue with SARS-CoV-2 virus nucleocapsid (green) (**G**–**I**). Merge staining for ocular tissue with SARS-CoV-2 virus nucleocapsid (green) and nuclear DAPI staining (blue) (**J**–**L**). Scale bars = 50 µM.

**Figure 3 jcm-11-03312-f003:**
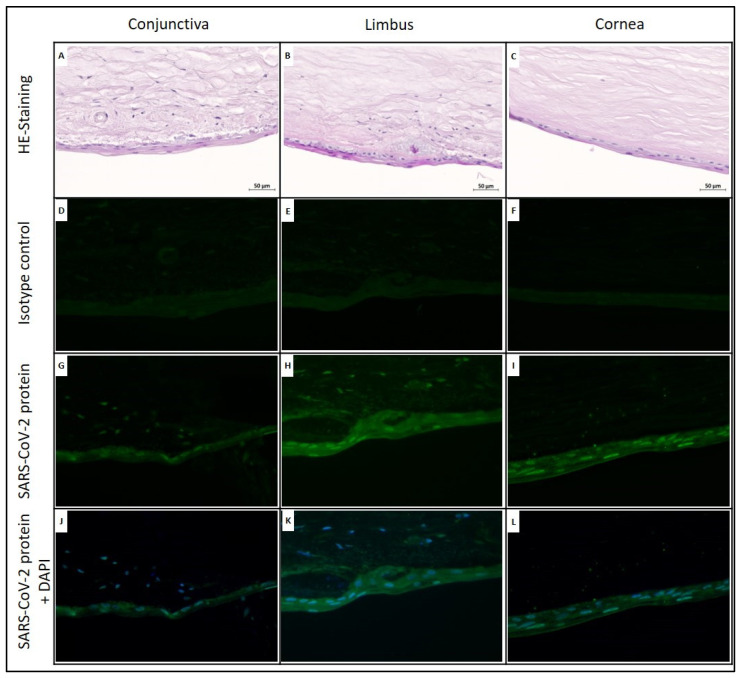
Immunohistochemical staining (Case 2, day 32). The SARS-CoV-2 Nucleocapsid protein was detected in the right cornea tissue of the COVID-19 donor (Case 2) after 32 days of in vitro cultivation; HE-staining of ocular tissues from conjunctiva, limbus and cornea (**A**–**C**). Isotype control of immunofluorescence analyses of ocular tissues (**D**–**F**). Immunofluorescence imaging of ocular tissue with the SARS-CoV-2 virus nucleocapsid (green) (**G**–**I**). Merge staining for ocular tissue with the SARS-CoV-2 virus nucleocapsid (green) and nuclear DAPI staining (blue) (**J**–**L**). Scale bars = 50 µM.

**Table 1 jcm-11-03312-t001:** Donor Characteristics.

Characteristics	Case 1	Case 2	Case 3	Case 4
** Age of death in years **	62	67	55	72
** Sex **	Male	Male	Female	Female
** Prior nonophthalmic history **	Severe mesenteric ischemia, pronounced generalized arteriosclerosis, therapy-resistant Crohn’s disease	Peripheral arterial occlusive disease with forefoot infection; death because of multimorbidity with decompensated heart failure and acute renal failure	Obesity, severe heart disease, diabetes and COPD	Stroke, arterial thrombus, arterial hypertension, diabetes mellitus and peripheral arterial occlusive disease
** SARS-CoV-2 vaccination **	No	No	No	No
** SARS-CoV-2 symptoms **	No	No	No	No
** Premortem lateral flow test **	Negative	Negative	Negative	Negative
** Latest premortem qRT–PCR test **	6 weeks before death (positive, CT > 30)	3 weeks before death (negative)	4 days before death (negative)	-
** Premortem serum antibody test for SARS-CoV-2 **	Negative for antibodies (IgA, IgG)	Negative for antibodies (IgG, IgG II)	Negative for antibodies (IgG, IgG II)	Negative for antibodies (IgG, IgG II)
** Postmortem conjunctival and nasopharyngeal swab test (qRT–PCR) for SARS-CoV-2 **	Positive (CT value of 20.7 (S and N gene, Anchor))	Positive (CT values of 21.7 (S and N gene, Anchor), 20.4 (E gene, Cepheid), 22.7 (N2 gene, Cepheid))	Positive (CT value of 16.2 (S and N gene, Anchor))	Positive (CT value of 39.2 (N and RdRP gene, Abbott))
** Strain **	Unknown	B.1.177	Unknown	Unknown
**Time of detection (days after death)**	4	7	1	4

COPD, chronic obstructive pulmonary disease; CT, cycle threshold.

**Table 2 jcm-11-03312-t002:** Overview of Time-Dependent qRT–PCR Results in Three Cases Using SARS-CoV-2 PCR (Target 1S and N-Gene, Anchor) during the Cultivation Time.

Sampling	Case 1		Case 2			Case 3			Case 4		
	Sample CS/CM	CT Value	Sample CS/CM	CT Value		Sample CS/CM	CT Value		Sample CS/CM	CT Value	
		Cornea Graft: L		Cornea Graft: R	Cornea Graft: L		Cornea Graft: R	Cornea Graft: L		Cornea Graft: R	Cornea Graft: L
2 d						CS	32.8	35.0			
						CM (o) ^b^	30.6	32.2			
5 d	CS ^a^	neg									
	CM (o) ^b^	38.0									
7 d	CS	neg	CS	34.3	34.3	CS	33.7	35.0	CS	neg	neg
	CM (o) ^b^	39.8	CM (n) ^b^	neg	neg	Li/Scl	31.6	33.4	CM (o) ^b^	neg	neg
	CM (n) ^b^	neg				Cornea	neg	neg			
						CM (o) ^b^	31.3	33.7			
14 d	CS	neg	CS	36.3	32.4	CS	32.4	33.4			
	CM (o) ^b^	neg	CM (n) ^b^	neg	neg	Li/Scl	35.9	34.1			
	CM (n) ^b^	neg				Cornea	37.2	neg			
						CM (o) ^b^	31.3	31.3			
						CM (n) ^b^	36.0	neg			
21 d			CS	35.9	neg	CS	33.9	36.5			
			Li/Scl	35.4	33.1	Li/Scl	33.2	37.8			
			Cornea	39.1	neg	Cornea	neg	neg			
						CM (o) ^b^	31.5	34.2			
						CM (n) ^b^	neg	neg			
28 d			^1^/_8_ CS	neg	35.1	^1^/_12_ CS	34.5	37.3			
			Li/Scl	35.2	neg	Li/Scl	34.5	39.2			
			Cornea	37.5	neg	Cornea	neg	neg			
						CM (o) ^b^	32.3	34.5			
						CM (n) ^b^	neg	neg			
32 d			^1^/_8_ CS fixed	-	-	^1^/_12_ CS fixed	-	-			

CS, circle segment (Figure 1); CM, culture medium; CT, cycle threshold; Li/Scl, limbus and sclera; L, left; neg, negative; n, new; R, right; o, old. ^a^ Sample undiluted: not evaluable, 1:10 dilution-negative. ^b^ “o”: old/CM before medium change, “n”: new/CM after medium change.

**Table 3 jcm-11-03312-t003:** Overview of RNA Expression and Subgenomic RNA Expression in Case 2.

Sampling		Sample	Detection ^a^ (CT Value)	N1 (CT Value)	sg N1 (CT Value)	Delta CT
7 d	R	^1^/_8_ CS	34.3	31.9	35.3	−3.4
	L	^1^/_8_ CS	34.3	32.3	36.4	−4.1
21 d	L	^1^/_8_ CS	-	-	-	-
		Li/Scl	33.1	31.6	35.6	−4.0
		Cornea	-	-	-	-
	R	^1^/_8_ CS	35.9	34.8	-	-
		Li/Scl	35.4	34.3	35.8	-
		Cornea	39.1	-	-	-
Cell culture supernatant (ctrl)				20.5	25.5	−5.2

CS, circle segment; CM, culture medium; CT, cycle threshold; Li/Scl, limbus and sclera (Figure 2).^a^ Detection with SARS-CoV-2 PCR (target 1S and N gene, Anchor), see Table 1.

**Table 4 jcm-11-03312-t004:** Overview of RNA Expression and Subgenomic RNA Expression in Case 3.

Sampling		Sample	Detection ^a^ (CT Value)	N1 (CT Value)	sg N1 (CT Value)	Delta CT
7 d	L	^1^/_12_ CS	35.0	32.4	36.3	−3.9
		Li/Scl	33.4	31.2	34.3	−3.1
		Cornea	-	-	-	-
	R	^1^/_12_ CS	33.7	31.2	36.3	−5.1
		Li/Scl	31.6	29.3	34.8	−5.5
		Cornea	-	37.2	-	-
14 d	L	^1^/_12_ CS	33.4	33.2	37.4	−4.2
		Li/Scl	34.1	32.9	-	-
		Cornea	-	-	-	-
		CM (o)	31.3	31.9	-	-
		CM (n)	-	-	-	-
	R	^1^/_12_ CS	32.4	32.2	36.5	−4.3
		Li/Scl	35.9 ^b^			
		Cornea	37.2	36.1	37.4	n.d. ^c^
		CM (o)	31.3	29.2	33.8	−4.6
		CM (n)	36.0	-	-	-
21 d	L	^1^/_12_ CS	36.5	32.9	-	-
		Li/Scl	37.8	34.3	-	-
		Cornea	-	-	-	-
	R	^1^/_12_ CS	33.9	29.6	35.6	−6
		Li/Scl	33.2	29.9	36.4	−6.5
		Cornea	-	36.8	-	-
Cell culture supernatant (ctrl)				20.5	25.7	−5.2

CM, culture medium; CS, circle segment; Li/Scl, limbus and sclera (Figure 2); n.d., not detected; n, new/after medium change; o, old/CM before medium change. ^a^ Detection with SARS-CoV-2 PCR (target 1S and N gene, Anchor), see Table 1. ^b^ No sample available for further examination. ^c^ Not detected, outside of linear range.

**Table 5 jcm-11-03312-t005:** Time-Dependent Overview of SARS-CoV-2 Immunofluorescence during the Cultivation Time.

Cultivation Time (d)	Ocular Tissue	IHC Staining of SARS-CoV-2 Protein	
		Cornea: R	Cornea: L
7	Conjunctiva	+	++
	Limbus	++	+
	Cornea	++	+
32	Conjunctiva	+	+++
	Limbus	+++	+
	Cornea	++	+

L, left; R, right; +, light staining; ++, moderate staining; +++, strong staining.

## Data Availability

Not applicable.

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
