# Peer review of "New Safety Aspects in Corneal Donation—Studies on SARS-CoV-2-Positive Corneal Donors"

_jcm, 2022, doi:10.3390/jcm11123312_

Round 1
Reviewer 1 Report
This study illustrates the presence of Coronavirus in tissue and culture medium albeit with a minimized transmission risk with prolongation of the culture period. It is very well conceptualized and conducted with a sound methodology. It is interesting to see that viral presence is more pronounced in the limbal/scleral area with a reduction towards the corneal center. There are a few minor points that need to be addressed which I note below.
1. Figure 2 is missing legend information for the images in panels G – L.
2. Please clarify what the connotations weak, medium and strong staining indicate in Table 5. In other words, is there a quantitative scale by which one can categorize the staining amongst these categories?
3. The authors have conducted analyses from a very wide perspective, however the aims section in the introduction is not clearly representative of what is following up in the manuscript. In its current form, this part of introduction is very weak in attracting attention to further read the following parts.
4. The findings of this manuscript is not only relevant in terms of the suitability of a Covid + donor cornea for transplantation but also gives insight into the assessment of keratolimbal donor suitability for other surgical operations (e.g. cadaveric keratolimbal allograft). This may need to be emphasized throughout the manuscript.
Author Response
We would like to thank you for the very helpful critical comments on the first draft of this manuscript.
We hope to improve the manuscript accordingly to your remarks.
- To 1: Figure 2 is missing legend information for the images in panels G – L.
We updated the legends of Figure 2 and 3 (line 255-258 and line 290-292) to add the information of panels G-L
- To 2: Please clarify what the connotations weak, medium and strong staining indicate in Table 5. In other words, is there a quantitative scale by which one can categorize the staining amongst these categories?
We added two sentences in the Methods section to explain how we evaluated the IHC stainings. Thereby, we changed “weak to “light” and “medium“ to “moderate” for a clearer wording. Line 155-157: “SARS-CoV-2 expression was evaluated by estimating a semiquantitative scoring system (light staining (+), moderate staining (++) and strong staining (***)). Evaluation was done by two experienced biologists.” In the results section, table 5 we updated the legends regarding these changes (line 308: “Table 5: L, left; R, right; +, light staining; ++, moderate staining; +++, strong staining”)
- To 3: The authors have conducted analyses from a very wide perspective, however the aims section in the introduction is not clearly representative of what is following up in the manuscript. In its current form, this part of introduction is very weak in attracting attention to further read the following parts.
In the aim section we tried to attract attention to further read the manuscript by rewriting this part (line 67-75): “Previous studies about SARS-CoV-2 infections in ocular tissues were mostly per-formed using ocular tissues from patients who died on SARS-CoV-2. In contrast, in this study, we investigated the presence of SARS-CoV-2 virus in ocular tissue from presumed negative tested cornea donors with post-mortem verified Covid-19 infection. Ocular tissue was analysed using qRT-PCR, subgenomic RNA analyses and immunohistochemical stainings. Thereby, we monitored the local and time-related distribution of SARS-CoV-2 virus during regular organ culture of corneas. Thus, the study presented here aims to determine whether there is an increased risk of transmission of SARS-CoV-2 from cornea donors.”
- To 4: The findings of this manuscript is not only relevant in terms of the suitability of a Covid + donor cornea for transplantation but also gives insight into the assessment of keratolimbal donor suitability for other surgical operations (e.g. cadaveric keratolimbal allograft). This may need to be emphasized throughout the manuscript.
In the discussion section, we added a part to the insights into the assessment of keratolimbal donor suitability for other surgical operations (line 448-451): “These findings are not only relevant in terms of the suitability of Covid posi-tive donor corneas for keratoplastics but also important for limbocorneal grafts and sclerocorneal grafts, because the cultured limbal region may contain active SARS-CoV2-virus.”
Furthermore, we improved some minor mistakes:
- Table 3: last line: “20 May” and “20 Jul” was changed in real numbers “5” and “20.7”, respectively.
- Table 4, last line: “positive control” was changed to “Cell culture supernatant (ctrl)”
- As an additional comment, we would like to inform you that this manuscript was proven for correct English language by the American Journal Experts (AJE) service.
Reviewer 2 Report
The authors have performed a very well designed study on the donor corneas for SARS-CoV-2. The results are very interesting , however it would be better to mention the patient's prior COVID history , Also there is not a control group in the study to confirm the value of the positive culture results, I suggest to add these two points as the weak points of the study.
Author Response
We performed further IHC stainings for SARS-CoV-2 on an enucleated eye from Covid-19 negative patient. The control was negative for SARS-CoV-2. In method section, we added this information in line (142-144): “At days 2, 7, and 28, a circle segment of the corneal grafts was used for IHC staining. As control tissue, enucleated eye from Covid-19 negative patient was added to the analyses. The tissues were…” The results were included in the results section (line 239 to 242): “As negative control, corneal tissue from a SARS-CoV-2 negative enucleated eye was used. IHC analyses showed no SARS-CoV-2 positive cells in the control (data not shown). In contrast, cytoplasmic SARS-CoV-2 staining was revealed in the epithelial cells of the right and left eye corneal circle segments after 7 days of cell culture (Figure 2).”
Furthermore, we improved some minor mistakes:
- Table 3: last line: “20 May” and “20 Jul” was changed in real numbers “5” and “20.7”, respectively.
- Table 4, last line: “positive control” was changed to “Cell culture supernatant (ctrl)”
- As an additional comment, we would like to inform you that this manuscript was proven for correct English language by the American Journal Experts (AJE) service.